# MAP Kinase Pathways in Brain Endothelial Cells and Crosstalk with Pericytes and Astrocytes Mediate Contrast-Induced Blood–Brain Barrier Disruption

**DOI:** 10.3390/pharmaceutics13081272

**Published:** 2021-08-17

**Authors:** Yuki Matsunaga, Shinsuke Nakagawa, Yoichi Morofuji, Shinya Dohgu, Daisuke Watanabe, Nobutaka Horie, Tsuyoshi Izumo, Masami Niwa, Fruzsina R. Walter, Ana Raquel Santa-Maria, Maria A. Deli, Takayuki Matsuo

**Affiliations:** 1Department of Neurosurgery, Graduate School of Biomedical Sciences, Nagasaki University, 1-7-1 Sakamoto, Nagasaki 852-8501, Japan; y.matsunaga923@gmail.com (Y.M.); nobhorie@nagasaki-u.ac.jp (N.H.); tizumo@nagasaki-u.ac.jp (T.I.); takayuki@nagasaki-u.ac.jp (T.M.); 2Department of Pharmaceutical Care and Health Sciences, Faculty of Pharmaceutical Sciences, Fukuoka University, 8-19-1 Nanakuma, Jonan-ku, Fukuoka 814-0180, Japan; shin3@fukuoka-u.ac.jp (S.N.); dohgu@fukuoka-u.ac.jp (S.D.); 3BBB Laboratory, PharmaCo-Cell Company Ltd., Dai-ichi-senshu bldg. 2nd Floor, 6-19 Chitose-machi, Nagasaki 852-8135, Japan; watanabe@pharmacocell.co.jp (D.W.); niwa@pharmacocell.co.jp (M.N.); 4Biological Barriers Research Group, Institute of Biophysics, Biological Research Centre, 6726 Szeged, Hungary; walter.fruzsina@brc.hu (F.R.W.); anaraquel.santamaria@wyss.harvard.edu (A.R.S.-M.); deli.maria@brc.hu (M.A.D.)

**Keywords:** blood–brain barrier, contrast media, iopamidol, MAP kinase, pericytes, astrocytes, neurointervention, contrast-induced encephalopathy

## Abstract

Neurointervention with contrast media (CM) has rapidly increased, but the impact of CM extravasation and the related side effects remain controversial. This study investigated the effect of CM on blood–brain barrier (BBB) integrity. We established in vitro BBB models using primary cultures of rat BBB-related cells. To assess the effects of CM on BBB functions, we evaluated transendothelial electrical resistance, permeability, and tight junction (TJ) protein expression using immunohistochemistry (IHC) and Western blotting. To investigate the mechanism of iopamidol-induced barrier dysfunction, the role of mitogen-activated protein (MAP) kinases in brain endothelial cells was examined. We assessed the effect of conditioned medium derived from astrocytes and pericytes under iopamidol treatment. Short-term iopamidol exposure on the luminal side induced transient, while on the abluminal side caused persistent BBB dysfunction. IHC and immunoblotting revealed CM decreased the expression of TJ proteins. Iopamidol-induced barrier dysfunction was improved via the regulation of MAP kinase pathways. Conditioned medium from CM-exposed pericytes or astrocytes lacks the ability to enhance barrier function. CM may cause BBB dysfunction. MAP kinase pathways in brain endothelial cells and the interactions of astrocytes and pericytes mediate iopamidol-induced barrier dysfunction. CM extravasation may have negative effects on clinical outcomes in patients.

## 1. Introduction

Intravascular injections of iodine contrast media (CM) are widely used in the field of radiology for diagnostic imaging, and CM has recently been utilized in endovascular interventions. In clinical practice, the intravascular administration concentration of iodine CM is often about 150–300 mgI/mL, and the CM is usually rapidly excreted in urine by renal glomerular filtration. Despite its widespread and increasing use in clinical settings, CM also has adverse effects that can result in temporary or permanent complications. The most well-known adverse effect of CM is contrast-induced nephropathy (CIN) [1]. It has been reported that the pathophysiology underlying CIN includes endothelial damage, renal tubular toxicity, and oxidative stress [2,3,4,5]. Contrast-induced encephalopathy (CIE), which comprises the neurological adverse effects of CM, is an uncommon complication of cerebral angiography with an estimated prevalence rate of 1–2% [6,7,8]. Patients with CIE may experience episodes of motor or sensory deficits, visual disturbance, aphasia, or seizures [7]. With the progress of endovascular intervention in recent years, interventions using CM such as coronary intervention and neurointervention have rapidly become widespread, and the incidence of CIE has reportedly increased [9]. In mechanical thrombectomy for acute ischemic stroke, in particular, postoperative imaging often confirms CM leakage into the brain parenchyma because of ischemia/reperfusion injury and repeated intra-arterial injections of CM [9]. Although chemical toxicity and osmotic pressure may be involved in the cause and mechanism of CIE, several unclear points remain, and no treatment policy has been established. It is considered that the breakdown of the blood–brain barrier (BBB), which regulates CM leakage into the brain parenchyma, is likely related to the induction of CIE. The BBB is essential for the maintenance and regulation of the central nervous system microenvironment. Brain capillary endothelial cells, pericytes, and astrocytes are the basic cellular components of the BBB. In addition to cellular components, BBB comprises the base membrane and extracellular matrix. The crosstalk between these cells is essential for the functional construction of the BBB. The collective concept for these cell types as a coordinated system in the brain is the neurovascular unit (NVU) [10,11,12]. To elucidate the pathophysiology of central nervous system disorder, it is necessary to understand the concept of NVU, which includes endothelial cells and TJ, as well as interactions with other neighboring cells. Deficiency in this crosstalk is involved in the development and progression of central nervous system diseases [13]. In vitro BBB models have been developed and used to evaluate BBB functions in many studies [14]. We can more closely recapitulate the in vivo biological environment to evaluate BBB functions by using a primary cell-based in vitro BBB model consisting of rat brain endothelial cells (RBECs), astrocytes, and pericytes [15]. In the present study, we investigated the effects of CM on primary isolated NVU cell types to elucidate the mechanism of contrast-induced BBB dysfunction using our well characterized in vitro BBB model.

## 2. Materials and Methods

### 2.1. Animals

Wistar rats were obtained from Japan SLC, Inc., Shizuoka, Japan. All animals were treated in strict accordance with the National Institutes of Health Guidelines for the Care and Use of Laboratory Animals (NIH Publication No. 80–23), and protocols were approved by the Nagasaki University animal care committee (Approval Number: 2003231610, 23 March 2020).

### 2.2. Materials and Reagents

All reagents were purchased from Sigma-Aldrich (St. Louis, MO, USA) unless indicated otherwise. Iopamidol was purchased from FUJIFILM Wako Pure Chemical Corporation (Osaka, Japan). The stock solution of iopamidol (612.4 mg/mL, iodine concentration of 300 mgI/mL) was prepared by dissolving it in an RBEC medium, as indicated below. The stock solution was diluted in RBEC medium to prepare the working solutions indicated in each experiment. Mannitol (62 mM) was used as an osmolarity control. The osmotic pressure of 62 mM mannitol is equivalent to that of 30 mgI/mL iopamidol.

### 2.3. Cell Culture

Primary RBECs were isolated from male Wistar rats (3–4 weeks old), as previously described [15,16,17,18]. RBECs were maintained in RBEC medium comprising Dulbecco’s modified Eagle’s medium (DMEM)/F12 supplemented with 10% fetal bovine serum (FBS; Biosera, France), basic fibroblast growth factor (FGF-2, 1.5 ng/mL), heparin (100 μg/mL), insulin (5 μg/mL), transferrin (5 μg/mL), sodium selenite (5 ng/mL) (insulin-transferrin-sodium selenite medium supplement), and gentamicin (50 μg/mL). For the first 2 days, cells were incubated in a medium containing puromycin (4 μg/mL) to avoid contamination from pericytes [19]. When the culture reached 80% confluency, purified endothelial cells were passaged and used for the experiments.

Rat cerebral astrocytes were obtained from neonatal Wister rats. Meninges were removed, and cortical pieces were mechanically dissociated in an astrocyte culture medium (DMEM supplemented with 10% FBS). Dissociated cells were seeded in cell culture flasks. To obtain type 1 astrocytes, flasks with confluent cultures were shaken at 37 °C overnight.

Rat cerebral pericytes were obtained via a prolonged, 2-week-long culture of isolated brain microvessel fragments consisting of pericytes and endothelial cells. The same preparations yielded primary RBECs after puromycin treatment. Pericyte survival and proliferation were favored by selective culture conditions using uncoated dishes and DMEM supplemented with 10% FBS and antibiotics. The culture medium was changed every 3 days. Astrocytes and pericytes were frozen in cryo-medium LABO Banker 2 (TOSC Japan Ltd., Tokyo, Japan), and stored at −80 °C until use.

The cell types used in the triple co-culture BBB model have been characterized in detail in our previous publications [15,18,20]. Brain endothelial cell cultures expressed endothelial characteristics [15] as well as BBB markers both at the gene and protein levels [15,21]. More than 90% of the cells in the astrocyte cultures stained positively for glial fibrillary acidic protein [15,18,20]. Pericyte cultures expressed NG-2 and alpha-smooth muscle actin [15,18,20]. The cell types have been regularly tested for these markers by immunocytochemistry.

### 2.4. Establishment of the In Vitro BBB Model

Two BBB models were used in this study: brain endothelial cells alone (E00) and endothelial cells in triple culture with astrocytes and pericytes (EPA). The E00 model was constructed using Millicell^®^ Hanging Cell Culture Inserts (0.4-μm pore size) in 24-well culture plates. RBECs (1.0 × 10^5^ cells/insert) were seeded on the upper side of the collagen- and fibronectin-coated membrane of the insert. To construct the EPA model, astrocytes (5.0 × 10^4^ cells/well) were seeded on the bottom of the collagen-coated 24-well culture plate, and pericytes (1.0 × 10^4^ cells/insert) were seeded on the bottom side of the collagen-coated membrane of the insert. Pericytes were allowed to adhere firmly overnight; then, the next day RBECs (1.0 × 10^5^ cells/insert) were seeded on the inserts, as described previously [15]. The BBB models were maintained in RBEC medium supplemented with 500 nM hydrocortisone [22].

### 2.5. Transendothelial Electrical Resistance (TEER)

To evaluate the barrier integrity of cultured brain endothelial cells, TEER was measured using an EVOM resistance meter (World Precision Instruments, Sarasota, FL, USA) and RBECs cultured on Millicell^®^ inserts in 24-well plates. The resistance values of blank filters without cells (background resistance) were subtracted from those of filters with cells. The values are presented as Ω × cm^2^, and data indicate the change in TEER measured before and after treatment as compared to the control group.

### 2.6. Paracellular Permeability of Sodium Fluorescein (Na-F)

The flux of Na-F through the BBB models was determined, as previously described [15,23,24,25,26]. Cell culture inserts were transferred to 24-well plates containing 0.9 mL of assay buffer (composition: 136 mM NaCl, 0.9 mM CaCl_2_, 0.5 mM MgCl_2_, 2.7 mM KCl, 1.5 mM KH_2_PO_4_, 10 mM NaH_2_PO_4_, 25 mM glucose, and 10 mM HEPES, in distilled water, sterile filtered, pH 7.4) in the basolateral compartment. In the inserts, the culture medium was replaced with 0.2 mL of buffer containing 10 μg/mL Na-F. Inserts were removed from the abluminal chamber after 30 min. The concentration of Na-F was determined using a fluorescence multi-well plate reader (Wallac 1420 ARVO Multilabel Counter, MA, USA) with emission at 535 nm and excitation at 485 nm. The flux of Na-F through the BBB models was used as an index of paracellular transport [16,26]. The apparent permeability coefficient (cm/s) was calculated as previously described [27].

### 2.7. CM Treatment in the In Vitro BBB Models

To examine the effect of short-term treatment with CM, iopamidol (300 mgI/mL) was added to the luminal (upper) side of the E00 model. Following 30 min of incubation, CM was replaced with RBEC medium. Then, barrier function was monitored by measuring TEER for 24 h. Permeability of Na-F through the E00 models was measured at 6 h after replacing treatment with RBEC medium. To examine the dose- and time-dependent effects of iopamidol on the barrier function, iopamidol (0, 3, 15, and 30 mgI/mL) was added to the abluminal (lower) side of the E00 model. As the hyperosmolarity of Iopamidol may affect BBB function, mannitol (62 mM), which has the same osmolality as iopamidol (30 mgI/mL), was used as an osmotic control. Then, barrier function was monitored by measuring TEER for up to 72 h. To examine the effect of fibroblast growth factor-2 (FGF-2, ReproCELL Inc., city, Japan) on barrier dysfunction induced by iopamidol, iopamidol (30 mgI/mL) and/or FGF-2 (10 ng/mL) for 24 h was added to the luminal side of the E00 model. The effects of iopamidol on pericytes and astrocytes were examined by using the E00 and EPA models. Iopamidol (30 mgI/mL) was added to the abluminal side of these models.

### 2.8. Collection of Conditioned Medium and Treatment in the In Vitro BBB Models

To examine the effects of iopamidol on astrocytes and pericytes, the conditioned medium from iopamidol-exposed cells was collected. Astrocytes or pericytes were cultured in 6 cm culture dishes in the RBEC medium. A cell-free dish was used as the negative control. After reaching 90% confluency, cells were incubated with iopamidol (15 mg/mL) for 36 h. Following centrifugation (300× *g*, 5 min), the conditioned medium collected from the supernatant and fresh RBEC medium were mixed in a 1:1 ratio (50% conditioned medium) and stored at −20 °C until use. The conditioned medium was administrated to the luminal and abluminal sides of the E00 model.

### 2.9. Cell Viability

NVU types were seeded into a 96-well plate (1.0 × 10^4^ cells/well). After treatment with iopamidol (0, 3, 15, 30 mgI/mL), mannitol (62 mM), or FGF-2 (0, 1, 5, 10 ng/mL) for 24 h, cell viability was determined using the Cell C9ount Kit (CCK)-8 assay in accordance with the manufacturer’s instructions (Dojindo Co., Kumamoto, Japan).

### 2.10. Immunostaining

To observe changes in interendothelial junctions, cells were stained with primary antibodies against ZO-1 junctional associated molecule (33-9100, Invitrogen, Carlsbad, CA, USA) and claudin-5 tight junction protein (35-2500, Invitrogen). All primary antibodies were used at a dilution of 1:50. As a secondary antibody, Alexa Fluor 488-conjugated donkey anti-mouse immunoglobulin (Invitrogen) was used at a dilution of 1:200. After 24 h of treatment with or without iopamidol (30 mgI/mL) or mannitol (62 mM), cultured cells were washed in phosphate-buffered saline (PBS) and fixed in 3% paraformaldehyde in PBS for 10 min. Following permeabilization with 0.1% Triton-X 100 for 10 min, cells were blocked with 3% bovine serum albumin–PBS for 30 min. Then, cells were incubated with anti-ZO-1 and anti-claudin-5 antibodies overnight at 4 °C. The cells were rinsed with PBS and incubated with the secondary antibody for 45 min at 37 °C. Preparations were mounted in Gel Mount (Biomeda, Foster City, CA, USA), and staining was examined under an EVOS^®^ FL Cell Imaging System (Thermo Fisher Scientific, Eugene, OR, USA).

### 2.11. Immunoblotting

To detect the proteins of interest, RBECs were treated with iopamidol (30 mgI/mL), mannitol (62 mM), FGF-2 (10 ng/mL), or 50% CM for the period indicated in each experiment. After treatment, RBECs were lysed in radioimmunoprecipitation assay buffer. After measuring the protein concentration of each sample using BCA protein assay reagent (Pierce, Rockford, IL, USA), the samples were separated on a 10% Tris-Glycine extended gel (Bio-Rad, Hercules, CA, USA). Proteins were transferred onto a PVDF membrane (Bio-Rad). After blocking with 3% bovine serum albumin in Tris-buffered saline, the membranes were incubated with antibodies overnight at 4 °C. Anti–claudin-5, anti-occludin (Invitrogen), and anti–ZO-1 mouse monoclonal antibodies were used at a dilution of 1:5000. Anti–VE-cadherin goat polyclonal antibody (Santa Cruz, Dallas, TX, USA) was used at a dilution of 1:4000. Anti–mitogen-activated protein (MAP) kinase (extracellular signal-regulated kinase [ERK]1/2, phosphor-ERK1/2, p38, phosphor-p38, c-Jun N-terminal kinase [JNK], phosphor-JNK) rabbit antibodies (Cell Signaling Technology, Danvers, MA, USA) were used at a dilution of 1:2500. Anti–β-actin mouse monoclonal antibody (Sigma, St. Louis, MO, USA, loading control) was used at a dilution of 1:10,000. To visualize the immunoreactive bands, blots were incubated in Clarity Max Western ECL substrate in accordance with the manufacturer’s instructions (Bio-Rad) and detected using the FluorChem SP Imaging System (Alpha Innotech Corp., San Leandro, CA, USA).

### 2.12. Statistical Analysis

All data are expressed as the mean ± standard error of the mean. Student’s *t*-test was employed to perform two-group comparisons, whereas analysis of variance, followed by Dunnett’s or Tukey–Kramer’s post hoc analysis was used for multiple-group comparisons (GraphPad Prism 8.4; GraphPad Software, San Diego, CA, USA). A *p*-value of less than 0.05 was considered statistically significant. All experiments were repeated at least twice, and the number of parallel samples in each experiment was 3–4.

## 3. Results

### 3.1. Effects of Short-Term Exposure to CM on Barrier Function

We examined the effect of an iodinated CM. Iopamidol is characterized as a high-osmolar type and contains one benzene ring in the structure; therefore, it is called a monomer. In clinical settings, iopamidol is injected at concentrations of 200–370 mgI/mL. To examine whether short-term exposure of CM affects the barrier integrity, in vitro BBB model (E00 model) was treated with iopamidol (300 mgI/mL) for 30 min. After replacing the CM with a fresh culture medium, TEER was measured. Short-term exposure to iopamidol decreased TEER, compared to the control value at 3, 6, and 12 h time points. This reduction of TEER induced by iopamidol exposure was recovered at 24 h (Figure 1a). Iopamidol also increased the paracellular flux of Na-F, a small water-soluble marker, across the RBEC monolayer at 6 h (Figure 1b). These findings suggested that short-term exposure to iopamidol might increase endothelial permeability and lead to the leakage of CM into the brain parenchyma.

Although intravascularly injected CM is rapidly eliminated from the body, CM can affect BBB function for a prolonged period if it leaks into the brain parenchyma for an extended period of time. Therefore, we examined the dose and time dependency of the effects of iopamidol on barrier function. Iopamidol (0, 3, 15, 30 mgI/mL) was administered to the abluminal (lower) side of the monolayer BBB model. Iopamidol (15, 30 mgI/mL) induced a significant decrease of TEER at 6 h, which was sustained for up to 72 h (Figure 2). Mannitol as osmotic control decreased TEER in the early phase (at 6 h), but the effect did not last for a long time (Figure 2). These findings indicate that iopamidol might induce barrier dysfunction independently of osmolarity.

### 3.2. Effects of Iopamidol on Cell Viability and Expression of TJ Proteins in RBECs

Iopamidol dose-dependently decreased the viability of RBECs measured by CCK-8 assay, whereas mannitol had no effect (Figure 3a). We also investigated whether barrier dysfunction induced by iopamidol is related to the expression change of TJ proteins with immunostaining and Western blotting. Immunostaining revealed that iopamidol (30 mgI/mL) treatment disrupted the continuous, pericellular belt-like pattern of junctional proteins ZO-1 and claudin-5 (Figure 3b). In addition, iopamidol (30 mgI/mL) treatment decreased the expression of the claudin-5 protein, while the expression of occludin, ZO-1, and VE-cadherin TJ proteins remained unchanged (Figure 3c).

### 3.3. Influence of Iopamidol on the Phosphorylation of MAP Kinases in RBECs

The MAP kinase pathway is activated by various stimuli, such as growth factors, cytokines, oxidative stress, and hyperosmotic stress. Activated MAP kinase affects cellular processes including gene expression, proliferation, and apoptosis. Previous reports indicated that iopamidol influenced the MAP kinase pathway in renal cells and tissues [28,29]. To investigate the possible mechanism of iopamidol-induced barrier dysfunction, expression levels of the total- and phosphorylated-MAP kinase were examined. Following treatment with iopamidol (30 mgI/mL) for 0, 3, and 6 h, phosphorylated MAP components (extracellular signal-regulated kinase 1 and 2 (ERK), c-Jun N-terminal kinase (JNK), p38 mitogen-activated protein kinase (p38)) were detected by Western blot. Iopamidol decreased phosphorylation of ERK and increased p38 phosphorylation in this study (Figure 4a), but there were no notable changes of MAP kinase expression in RBECs treated with mannitol having the same osmolarity (Figure 4b).

### 3.4. Effects of FGF-2 on Viability and Barrier Function in Iopamidol-Treated RBECs

Based on the change of MAP kinase activation induced by iopamidol treatment, we next examined the effects of the regulation of MAP kinase activation on BBB function. It is known that several growth factors regulate the MAP kinase pathway. We examined the effect of FGF-2 on ERK and p38 phosphorylation. FGF-2 (10 ng/mL) treatment for 6 h increased ERK phosphorylation and decreased p38 phosphorylation (Figure 5a). As FGF-2 has a counter-regulatory effect on MAP kinase activation as iopamidol, we examined the effect of FGF-2 on iopamidol-induced BBB dysfunction. To examine the effect of FGF-2 on cell viability in iopamidol-treated RBECs, the cells were treated with iopamidol (15, 30 mgI/mL), in combination with FGF-2 (0–10 ng/mL) luminally for 24 h. Cell viability decreased by Iopamidol was dose-dependently enhanced by FGF-2 treatment (Figure 5b). We also examined the effect of FGF-2 on barrier dysfunction induced by iopamidol. RBECs (E00 model) were treated with iopamidol (30 mgI/mL), with or without FGF-2 (10 ng/mL) for 24 h. FGF-2 partially improved the iopamidol-induced barrier disruption (Figure 5c,d). These findings suggested that iopamidol disrupted the barrier integrity of brain endothelial cells through the MAP kinase pathways.

### 3.5. Effects of Iopamidol on NVU Types

The properties of the BBB are induced and maintained by crosstalk between brain capillary endothelial cells and the neighboring cells along the brain capillary, such as astrocytes and pericytes. Extravasated iopamidol may affect these cells and brain capillary endothelial cells. We hypothesized that iopamidol affects astrocytes and/or pericytes leading to abnormal interaction among these three cell types, which may disturb BBB function. To compare the effect of iopamidol on barrier function between the endothelial monolayer model (E00) and the endothelial–pericyte–astrocyte co-culture model (EPA), the abluminal side of these BBB models were treated with iopamidol (30 mgI/mL) for 24 h. Iopamidol decreased TEER (Actual TEER values; E00 control: 255.2 ± 6.62 (Ω × cm^2^), E00 iopamidol: 123.7 ± 9.92 (Ω × cm^2^), EPA control: 439.7 ± 29.41 (Ω × cm^2^), EPA iopamidol: 131.9 ± 9.26 (Ω × cm^2^)) and increased the permeability of the BBB to Na-F in both the E00 and EPA models. However, the size of the effect relative to the control was larger in the EPA model (Figure 6a,b). This finding suggested that iopamidol directly affects the activities of astrocytes and/or pericytes, thereby enhancing iopamidol-induced BBB damage. Next, we investigated the effect of iopamidol on the viability of astrocytes and pericytes. Iopamidol dose-dependently decreased the viability in both cell types (Figure 6c,d).

### 3.6. Effects of Iopamidol on the Astrocyte- or Pericyte-Induced Enhancement of Barrier Function in RBECs

It is known that astrocyte- and pericyte-derived humoral factors enhance the properties of the BBB [11]. As iopamidol influenced the viability of astrocytes and pericytes, humoral factors derived from iopamidol-exposed cells may be changed compared to untreated cells. Therefore, we investigated the effects of conditioned medium samples collected from iopamidol (15 mgI/mL)-treated or untreated (control) cells. The collected conditioned medium samples were diluted 1:1 with fresh RBEC medium (50% conditioned medium). Conditioned medium derived from astrocytes and pericytes under control conditions increased barrier functions. In contrast, conditioned medium derived from astrocytes and pericytes exposed to iopamidol did not enhance barrier function (Figure 7a,b). Western blotting revealed increased claudin-5 expression in the control groups, which is consistent with enhancement of barrier function as measured by TEER (Figure 7c).

## 4. Discussion

In the present study, we investigated the effects of the contrast agent iopamidol on culture models of the BBB and individual cell types forming the BBB. The results revealed that short-term exposure to clinically relevant concentrations of iopamidol on the luminal side induced BBB dysfunction—a change that was reversible. This temporary BBB dysfunction might allow the leakage of CM into the brain parenchyma. The presence of low concentrations of iopamidol in the brain parenchyma (abluminal side) can lead to permanent BBB disruption. We found that iopamidol worsened BBB function via the MAP kinase pathways (ERK pathway downregulation and p38 pathway activation). In addition, iopamidol promoted BBB injury through direct effects on pericytes and astrocytes.

Iopamidol is a water-soluble compound that rarely crosses the BBB. After an intravascular injection, iopamidol is promptly excreted through the urine via renal function, and the concentration of CM in blood also tends to decrease rapidly. However, there are clinical situations in which brain endothelial cells are exposed to high concentrations of iopamidol because of delayed excretion associated with renal dysfunction, repeated intra-arterial injection (e.g., neurointervention), and cerebral circulatory insufficiency. Renal failure, prior stroke, and heart failure have been reported as risk factors for CIE [9], which is consistent with these situations. CM exposure temporarily impairs BBB function, allowing CM to pass through the BBB into the brain parenchyma. This condition is also clinically observed, and computed tomography after mechanical thrombectomy for acute ischemic stroke can reveal CM leakage that matches the CM injection area [30]. In addition, iopamidol damaged endothelial cells of the rat aorta in vivo and decreased human umbilical vein endothelial cell (HUVEC) and H5V cell viability in vitro measured by CCK8 assay [31]. Although there are some reports of CM-induced cytotoxicity in other organs (e.g., human vascular smooth muscle cells, leukocytes, mast cells [32,33,34]), there are no reports indicating that iopamidol damages brain endothelial cells and causes BBB dysfunction. The present findings suggest that short-term exposure to a high concentration of iopamidol causes temporary BBB injury. Iopamidol that leaked into the abluminal side (representing the side of brain parenchyma), even at a low concentration, decreased endothelial cell viability and TJ protein expression, leading to lasting BBB dysfunction. The effect of osmotic pressure is considered one of the drivers of CM-induced endothelial cell damage. However, our experiments revealed that the decrease in RBEC viability, TJ protein expression, and BBB function was independent of the osmotic pressure alone, although a temporary decrease in TEER was observed by mannitol used at the same osmolarity.

MAP kinases are serine–threonine kinases that mediate intracellular signaling associated with a variety of cellular activities, including cell proliferation, differentiation, survival, death, and transformation [35,36]. The main MAP kinase family consists of ERK, p38, and JNK pathways. The ERK pathway, which is the classical MAP kinase pathway, is activated by growth factors, and it acts mainly on cell proliferation. The p38 and JNK pathways, which are stress response MAP kinase pathways, are induced by environmental stress, osmotic shock, UV irradiation, oxidative stress, protein synthesis inhibitors, and pro-inflammatory cytokines, and they play a central role in inducing apoptosis and controlling immune responses and inflammation [35]. Previous reports indicated that CM-induced renal apoptosis may be mediated by MAP kinase signaling, especially the JNK/p38 MAP kinase pathway [28]. However, the mechanism by which iopamidol affects brain endothelial cells is unclear. It has also been reported that CM reduces proliferation and increases apoptosis in HUVECs [32]. Our results suggested that iopamidol activated the MAPK pathway, suppressed brain endothelial cell proliferation by downregulating the ERK pathway, and induced inflammation and apoptosis by activating the p38 pathway, resulting in decreased BBB function. No prior studies indicated that the p38 pathway mediated the iopamidol-induced BBB dysfunction. The MAP kinase pathway may also be a therapeutic target for contrast-induced BBB dysfunction.

Brain endothelial cells have dynamic interactions with other neighboring cells. Crosstalk between the cells of the NVU is crucial for the formation and maintenance of a functional BBB [11,15]. CM extravasation may affect these cells, as well as brain capillary endothelial cells. No previous studies to date described the relationships of CM with pericytes and/or astrocytes. Our experiments show that iopamidol can dose-dependently decrease the viability of both astrocytes and pericytes, and conditioned medium derived from astrocytes and pericytes exposed to iopamidol lacked the ability to enhance barrier function in RBECs. A possible explanation is that iopamidol decreases the viability of astrocytes and pericytes, resulting in BBB dysfunction because of reduced secretion of humoral factors that act protectively for the BBB. Iopamidol affects the activities of astrocytes and/or pericytes via humoral factors and enhances iopamidol-induced BBB damage.

In conclusion, short-term exposure to iopamidol induces temporary BBB dysfunction. In addition, low concentrations of iopamidol in the brain parenchyma (abluminal side) leads to lasting BBB disruption. Our present study illustrated that iopamidol worsened BBB function via the MAP kinase pathway (ERK pathway downregulation and p38 pathway activation). In addition, iopamidol also acts on pericytes and astrocytes to promote BBB injury. Iopamidol-induced BBB disruption may be related to the pathophysiology of the CIE. CM extravasation may have negative effects on the clinical outcomes in patients.

## Figures and Tables

**Figure 1 pharmaceutics-13-01272-f001:**
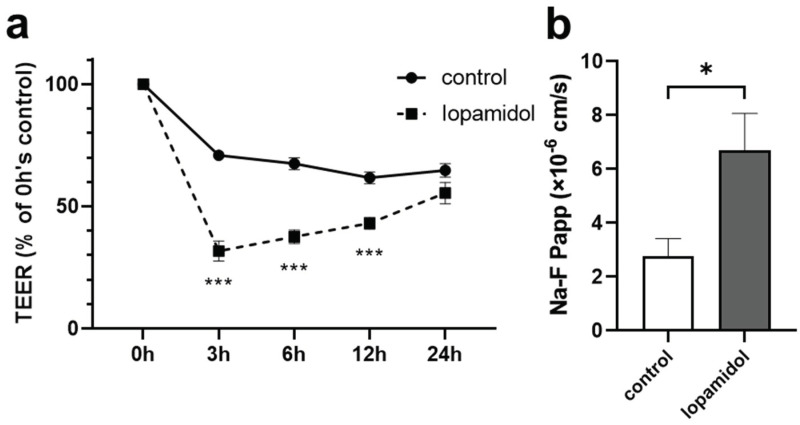
Effects of short-term exposure to contrast media on barrier functions using in vitro blood–brain barrier (BBB) models. Monolayer BBB models were exposed to iopamidol (300 mgI/mL) for 30 min. After replacing contrast media with RBEC medium, recovery of barrier function was evaluated by measuring transendothelial resistance (TEER) and the permeability of the BBB to sodium fluorescein (Na-F): (**a**) iopamidol decreased TEER at 3, 6, and 12 h. The reduction of TEER induced by iopamidol exposure was recovered at 24 h; (**b**) the permeability of the BBB to Na-F was examined at 6 h. Iopamidol increased the permeability of Na-F (*n* = 4, unpaired *t*-test, * *p* < 0.05, *** *p* < 0.001).

**Figure 2 pharmaceutics-13-01272-f002:**
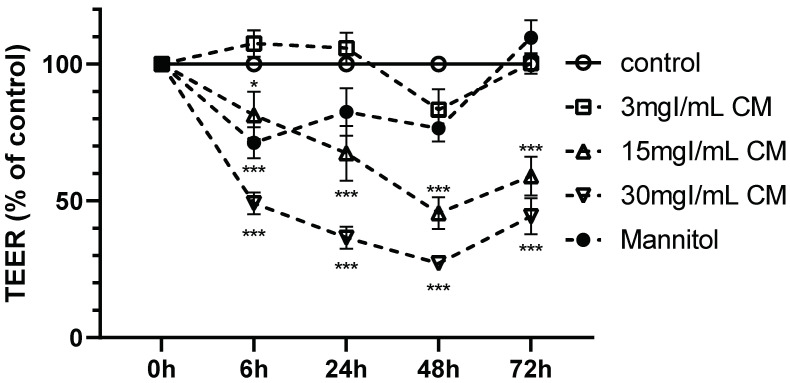
Dose and time dependency of the effects of iopamidol treatment on barrier function in the in vitro blood–brain barrier (BBB) models. Monolayer BBB models were exposed to iopamidol (0, 3, 15, 30 mgI/mL) abluminally for up to 72 h. Mannitol (62 mM) was used as an osmolarity control for iopamidol (30 mgI/mL). Barrier integrity was assessed by transendothelial resistance (TEER) measurement. Iopamidol (15 and 30 mgI/mL) significantly decreased TEER at 6 h, and the effect was sustained for up to 72 h. TEER is expressed as a percent of the control value at each time point (*n* = 4, one-way ANOVA, *p*-values were calculated from Tukey’s multiple comparison test, * *p* < 0.05, *** *p* < 0.001). CM, contrast media.

**Figure 3 pharmaceutics-13-01272-f003:**
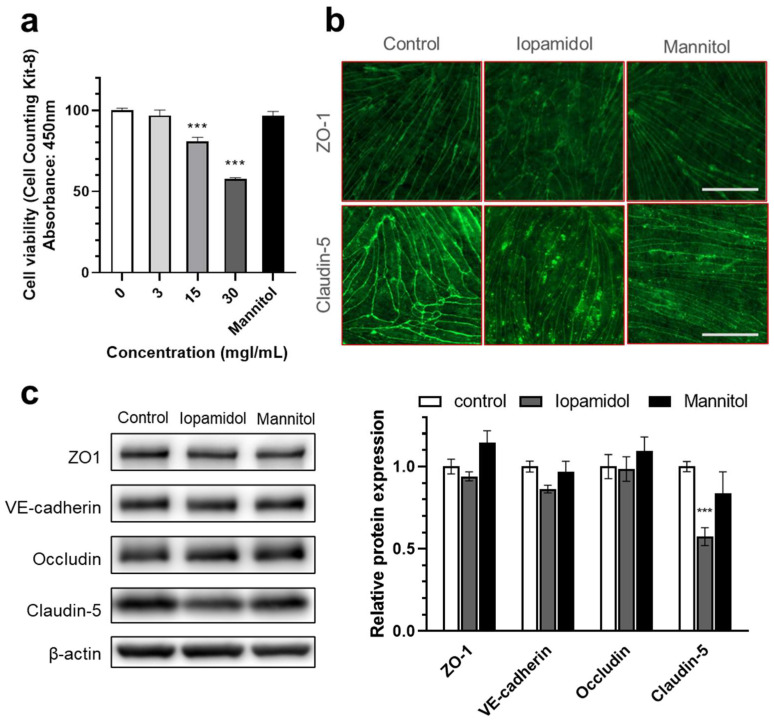
Effects of iopamidol on viability and tight junction protein expression in rat brain endothelial cells (RBECs): (**a**) RBECs were treated with iopamidol (0–30 mgI/mL) for 24 h. Cell viability was evaluated using the Cell Counting Kit-8 assay (*n* = 4–8). Iopamidol dose-dependently decreased cell viability. To examine the expression of tight junction proteins by immunostaining and Western blotting, RBECs were treated with iopamidol (30 mgI/mL) for 24 h (one-way ANOVA, p-values were calculated from Tukey’s multiple comparison test, *** *p* < 0.001); (**b**) iopamidol (30 mgI/mL) treatment disrupted the continuous pericellular pattern of tight junction proteins ZO1 and claudin-5. Bar = 50 μm; (**c**) representative images of Western blot experiments and relative expression of tight junction proteins (*n* = 7). Iopamidol (30 mgI/mL) decreased the expression of claudin-5 (one-way ANOVA, *p*-values were calculated from Tukey’s multiple comparison test, *** *p* < 0.001).

**Figure 4 pharmaceutics-13-01272-f004:**
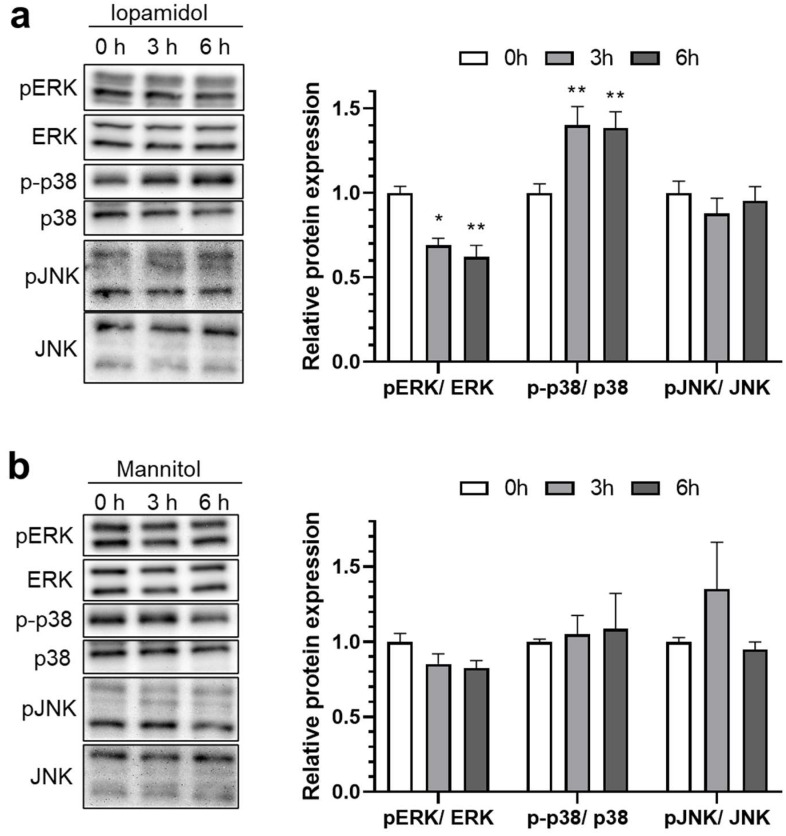
Effects of iopamidol on the expression of mitogen-activated protein (MAP) kinases. Rat brain endothelial cells (RBECs) were treated with iopamidol (30 mgI/mL) or mannitol (62 mM) for 0, 3, or 6 h. Total or phosphorylated MAP kinases (extracellular signal-regulated kinase [ERK], p38, and c-Jun N-terminal kinase [JNK]) were detected by Western blotting: (**a**) representative images of the Western blot. The graph presents the relative expression of MAP kinases in iopamidol-treated RBECs (*n* = 4–5, one-way ANOVA, *p*-values were calculated from Tukey’s multiple comparison test, * *p* < 0.05, ** *p* < 0.01). Iopamidol decreased ERK phosphorylation and increased p38 phosphorylation; (**b**) representative images of the Western blot. The graph presents the relative expression of MAP kinases in mannitol-treated RBECs (*n* = 3, one-way ANOVA, *p*-values were calculated from Tukey’s multiple comparison test). A significant change in phosphorylation levels in MAP kinases was not detected in mannitol-treated RBECs.

**Figure 5 pharmaceutics-13-01272-f005:**
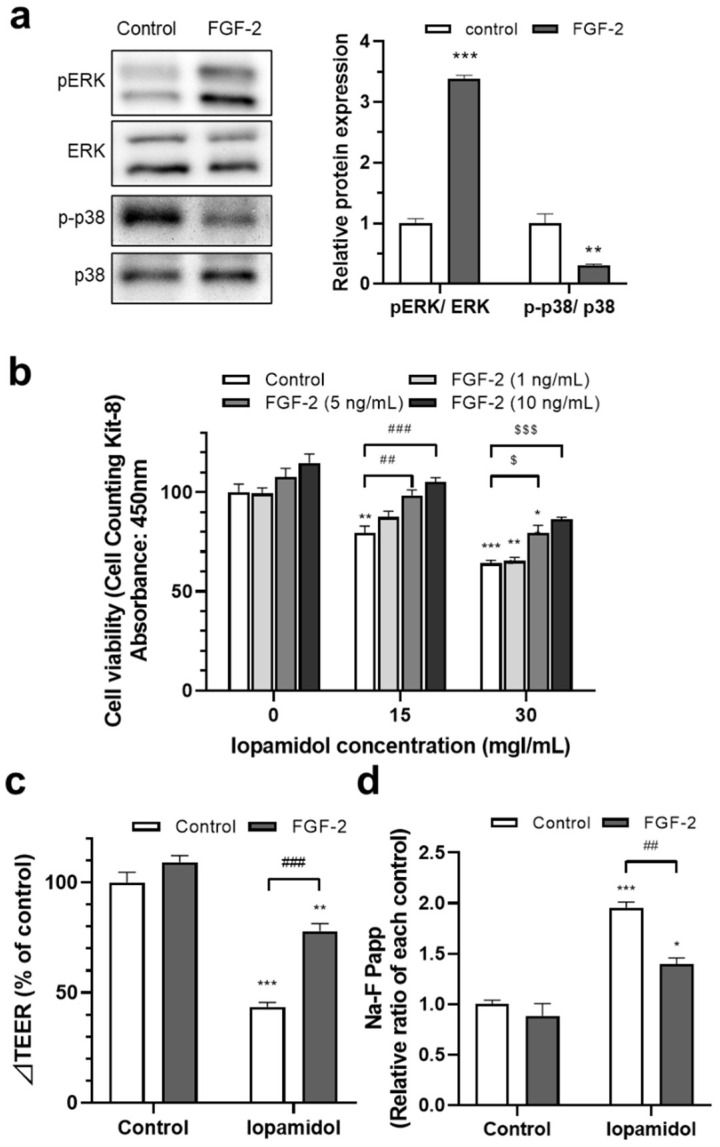
Effects of fibroblast growth factor-2 (FGF-2) on the viability and barrier function in iopamidol-treated rat brain endothelial cells (RBECs): (**a**) RBECs were treated with FGF-2 (10 ng/mL) for 6 h. Total or phosphorylated mitogen-activated protein (MAP) kinase expression (extracellular signal-regulated kinase [ERK] and p38) was detected by Western blotting. The blots are representative images of the experiment. The graph presents the relative expression of MAP kinases (*n* = 3, unpaired *t*-test, ** *p* < 0.01, *** *p* < 0.001); (**b**) to examine the effect of FGF-2 on viability in iopamidol-treated RBECs, cells were treated with FGF-2 (0–10 ng/mL) and iopamidol (15 and 30 mgI/mL) for 24 h (*n* = 4, two-way ANOVA, *p*-values were calculated from Dunnett’s multiple comparison test, * *p* < 0.05, ** *p* < 0.01, *** *p* < 0.001 vs. control, ^##^
*p* < 0.01, ^###^
*p* < 0.001, ^$^
*p* < 0.05, ^$$$^
*p* < 0.001). Iopamidol-induced reductions of cell viability were dose-dependently restored by FGF-2 treatment. To examine the effect of FGF-2 on barrier dysfunction induced by iopamidol, an in vitro BBB model was treated with iopamidol (30 mgI/mL) with FGF-2 (10 ng/mL) for 24 h. Barrier function was evaluated by TEER (**c**) and the permeability of the BBB to Na-F (**d**). FGF-2 partially reversed iopamidol-induced barrier dysfunction (*n* = 3–4, two-way ANOVA, *p*-values were calculated from Dunnett’s multiple comparison test, * *p* < 0.05, *** *p* < 0.001 vs. control, ^##^
*p* < 0.01).

**Figure 6 pharmaceutics-13-01272-f006:**
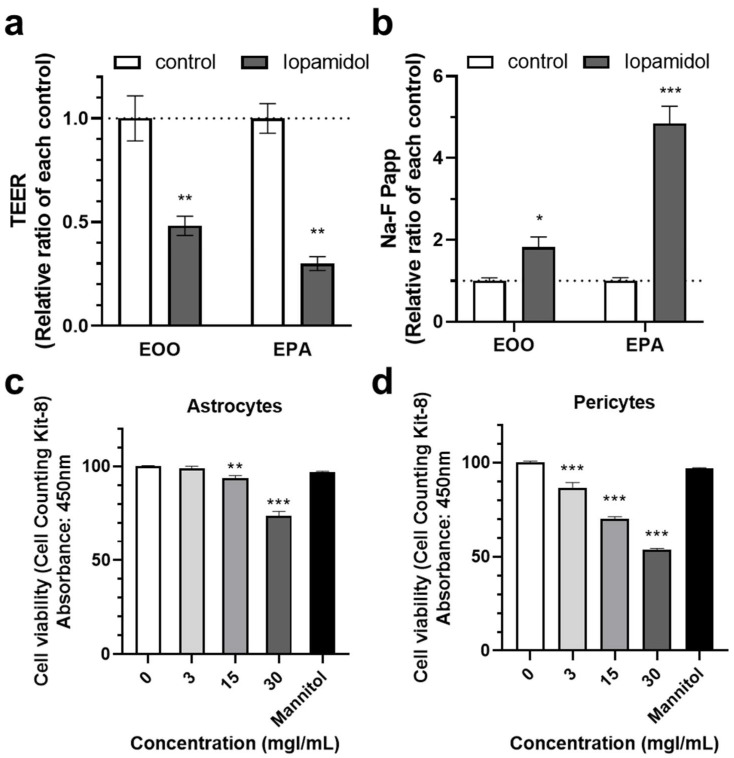
Effects of iopamidol on pericytes and astrocytes. Iopamidol (30 mgI/mL) was added to the abluminal side of the monolayer or co-culture blood–brain barrier (BBB) models for 24 h. Barrier function was evaluated by transendothelial electrical resistance (TEER) (**a**) and the permeability of the BBB to Na-F (**b**) (*n* = 3–4, unpaired *t*-test, * *p* < 0.05, ** *p* < 0.01, *** *p* < 0.001). To examine the effect of iopamidol on cell viability, astrocytes (**c**) and pericytes (**d**) cells were treated with iopamidol (0–30 mgI/mL) for 24 h. Iopamidol decreased the viability of astrocytes (*n* = 8) and pericytes (*n* = 4–8) (one-way ANOVA, *p*-values were calculated from Tukey’s multiple comparison test, ** *p* < 0.01, *** *p* < 0.001).

**Figure 7 pharmaceutics-13-01272-f007:**
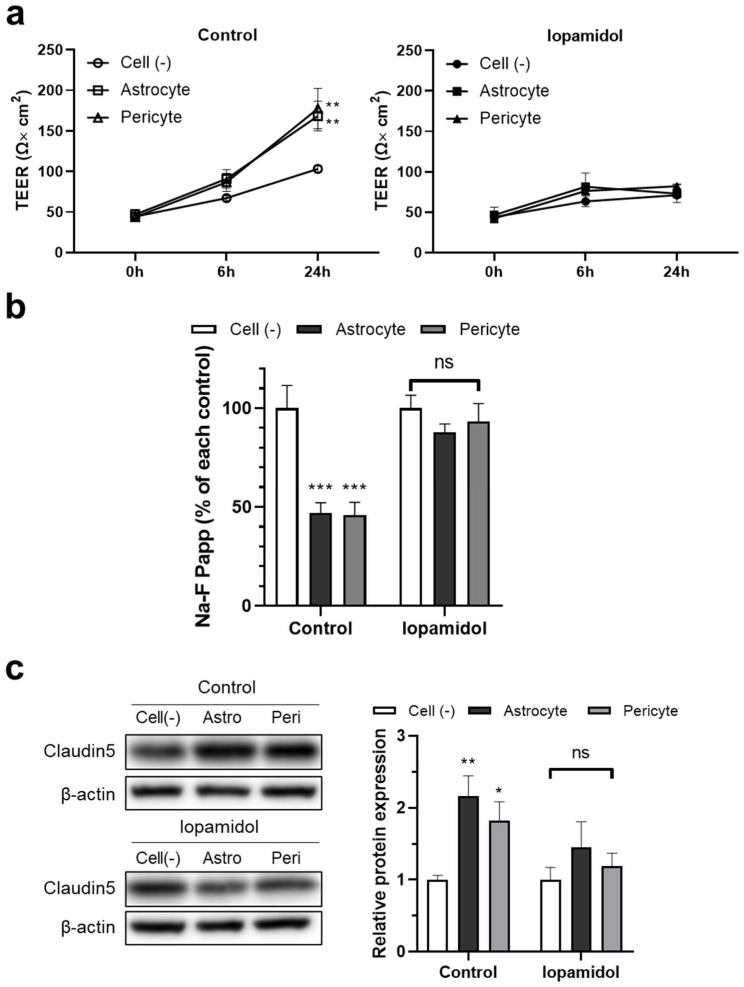
Effects of iopamidol on the astrocyte- or pericyte-induced enhancement of barrier function in rat brain endothelial cells (RBECs). To collect the conditioned medium, cell-free medium, astrocytes, or pericytes were incubated with iopamidol (15 mg/mL) for 36 h. The collected conditioned medium was diluted by 50% with fresh RBEC medium and used to treat RBECs monolayer model for 24 h. Barrier function was evaluated by transendothelial electrical resistance (TEER) (**a**) and the permeability of the blood–brain barrier (BBB) model to Na-F (**b**). The ability of astrocytes or pericytes to strengthen BBB barrier function was diminished by exposure to iopamidol. (**c**) Effect of conditioned medium on claudin-5 expression as examined by Western blotting. The blots are representative images of the experiment. The graph represents the relative expression of claudin-5 (*n* = 4, two-way ANOVA, *p*-values were calculated from Dunnett’s multiple comparison test, * *p* < 0.05, ** *p* < 0.01, *** *p* < 0.001, ns = no statistically significant differences).

## Data Availability

The data presented in this study are available on request from the corresponding author.

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
