# Peer review of "MAP Kinase Pathways in Brain Endothelial Cells and Crosstalk with Pericytes and Astrocytes Mediate Contrast-Induced Blood–Brain Barrier Disruption"

_pharmaceutics, 2021, doi:10.3390/pharmaceutics13081272_

Round 1
Reviewer 1 Report
In this study triple cultures of primary rat brain endothelial cells, pericytes and astrocytes were grown in an in vitro transwell system to mimic the neurovascular unit and the effects of a non-ionic iodinated contrast media (CM), iopamidol at clinically used concentrations was studied on the blood-brain barrier integrity. Several readout such as TEER, endothelial cell permeability and TJ protein expression was studied using IHC and WB analysis.
Minor points:
- There are several typos throughout the text and figures. A spell check is recommended. For example in Fig 6, y-axis should be control. line 375: Through.
- Fig 1a: Correct Y-axis (% of ohm). How was the data analyzed?
- Figure legends should capture the statistical test used to analyze data
- The actual TEER values in EOO and EPA cultures should be shown.
Major points:
- A paragraph on the neurovascular unit and blood-brain barrier should be included in the introduction.
- Astrocyte end-feet are main components of the neurovascular unit. The effects of iopamidol in a culture system where astrocytes and pericytes are seeded on the abluminal side of the filter should have been included. This model will recapitulate cellular interactions.
Author Response
Response to Reviewer 1 Comments
We wish to express our appreciation to the Reviewer for his or her insightful comments, which have helped us significantly improve the paper.
In this study triple cultures of primary rat brain endothelial cells, pericytes and astrocytes were grown in an in vitro transwell system to mimic the neurovascular unit and the effects of a non-ionic iodinated contrast media (CM), iopamidol at clinically used concentrations was studied on the blood-brain barrier integrity. Several readout such as TEER, endothelial cell permeability and TJ protein expression was studied using IHC and WB analysis.
Response: Thank you for your favorable comments.
Minor points:
- There are several typos throughout the text and figures. A spell check is recommended. For example in Fig 6, y-axis should be control. line 375: Through.
Response 1: Thank you for your advice. We have done a spell check and corrected some typos.
- Fig 1a: Correct Y-axis (% of ohm). How was the data analyzed?
Response 2: The Y-axis in Fig. 1 shows the percentage based on the control TEER just before the experiment (0h). The Y-axis in other figures is based on the control TEER at each time.
- Figure legends should capture the statistical test used to analyze data.
Response 3: We have added the statistical test used to analyze data in the figure legends according to your suggestions.
- The actual TEER values in EOO and EPA cultures should be shown.
Response 4: In accordance with the reviewer's comment, we have added the actual TEER value in E00 and EPA model in the manuscript (p11, line337-339)
Major points:
- A paragraph on the neurovascular unit and blood-brain barrier should be included in the introduction.
Response 1: Thank you for your suggestions. We agree with your suggestions. According to your suggestion, A paragraph on the neurovascular unit and BBB has been included in the introduction. (p2, line62-68)
- Astrocyte end-feet are main components of the neurovascular unit. The effects of iopamidol in a culture system where astrocytes and pericytes are seeded on the abluminal side of the filter should have been included. This model will recapitulate cellular interactions.
Response 2: Suggested studies on astrocyte end-feet are interesting and important; we will focus on them in future paper. One of the focuses for this paper was mainly on the crosstalk between astrocytes and pericytes from the perspective of humoral factors.

Reviewer 2 Report
The manuscript presents interesting data related to the molecular mechanisms involved in crosstalk of brain endothelial cells and pericytes and astrocytes when are subjected to treatment with contrast media.
For the results to be conclusive please demonstrate that the isolated cells are astrocytes, pericytes and endothelial cells. Some flow cytometry would sustaine the results.
Author Response
Response to Reviewer 2 Comments
We wish to express our appreciation to the Reviewer for insightful comments, which have helped us significantly improve the paper.
The manuscript presents interesting data related to the molecular mechanisms involved in crosstalk of brain endothelial cells and pericytes and astrocytes when are subjected to treatment with contrast media.
Response: Thank you for your favorable comments.
For the results to be conclusive please demonstrate that the isolated cells are astrocytes, pericytes and endothelial cells. Some flow cytometry would sustaine the results.
Response: I agree with the reviewer’s comment about the accuracy of the isolated cells. Our research group has previously published papers on primary cell-based in vitro BBB models consisting rat brain endothelial cells, astrocytes, and pericytes. In the present experiment, isolated cells were collected by the same culture methods. These details are provided in reference (DOI: 10.1016/j.neuint.2008.12.002, 10.1007/s10571-007-9195-4). We have modified the manuscript (p3, line111-117) and added the reference about isolated cells, characterized using the same culture method (DOI: 10.1016/j.snb.2015.07.110, 10.1186/s12987-019-0166-1, 10.3389/fnmol.2018.00166).

Reviewer 3 Report
In this manuscript the authors investigated the effect of widely used diagnostic contrast media (CM) (iopamidol) on the BBB integrity. The study is well-planned and well-described, the methods and results are clearly discussed. The molecular background of the dangerous effect of the CM on the brain microvascular permeability is also analysed. The role of MAP kinase pathway is elucidated in barrier dysfunction after iopamidol treatment.
The findings of the study are novel, important and call attention on the safety concerns when these CM-s are applied for diagnostic purposes.
Comments:
In the introduction a paragraph should be added about the dosage of the CM applied intravascularly in human radiology.
Some explanation is missing about the iopamidol doses/concentrations applied in the in vitro studies. Do they have human relevancy?
A sentence about the age-associated increasing BBB extravasation would also be interesting.
A review article summarizing the different cellular and molecular factors playing a role in BBB integrity should be added to the text and list of references: doi: 10.1177/0271678X16679420
Author Response
Response to Reviewer 3 Comments
We wish to express our appreciation to the Reviewer for his or her insightful comments, which have helped us significantly improve the paper.
In this manuscript the authors investigated the effect of widely used diagnostic contrast media (CM) (iopamidol) on the BBB integrity. The study is well-planned and well-described, the methods and results are clearly discussed. The molecular background of the dangerous effect of the CM on the brain microvascular permeability is also analysed. The role of MAP kinase pathway is elucidated in barrier dysfunction after iopamidol treatment.
The findings of the study are novel, important and call attention on the safety concerns when these CM-s are applied for diagnostic purposes.
Response: Thank you for your favorable comments.
Comments:
Point 1. In the introduction a paragraph should be added about the dosage of the CM applied intravascularly in human radiology.
Response 1: According to your suggestion, we have added the explanation of the dosage of the CM in clinical practice (p1, line41-43).
Point 2. Some explanation is missing about the iopamidol doses/concentrations applied in the in vitro studies. Do they have human relevancy?
Response 2: We have added the concentrations of the iopamidol, mannitol, and FGF-2 applied in the vitro studies (p5, line175, 193-194).
The renal function usually excretes CM without delay. But CM exposure and other factors (repeat injection, ischemia, renal failure etc.) cause temporary BBB opening and extravasation of CM. We think that stagnation of low dosage of the CM in the abluminal side worsens permanent BBB function. In clinical practice, CM extravasation after endovascular therapy, especially after mechanical thrombectomy for acute ischemic stroke, is observed on CT imaging. We believe it is quite possible that low concentrations of CM will remain on the abluminal side.
Point 3. A sentence about the age-associated increasing BBB extravasation would also be interesting.
Response 3: Thank you for your suggestion. We totally agree with your interest between aging and BBB. Suggested interests for age-associated BBB changes are very exciting and important. We rarely mention age-related changes in this paper, so we will focus on them in future papers.
Point 4. A review article summarizing the different cellular and molecular factors playing a role in BBB integrity should be added to the text and list of references: doi: 10.1177/0271678X16679420
Response 4: Thank you for your suggestion. We have added the reference suggested review (DOI: 10.1016/j.snb.2015.07.110) to strengthen the BBB integrity explanations.
